# Deep Learning-Based Microscopic Diagnosis of Odontogenic Keratocysts and Non-Keratocysts in Haematoxylin and Eosin-Stained Incisional Biopsies

**DOI:** 10.3390/diagnostics11122184

**Published:** 2021-11-24

**Authors:** Roopa S. Rao, Divya B. Shivanna, Kirti S. Mahadevpur, Sinchana G. Shivaramegowda, Spoorthi Prakash, Surendra Lakshminarayana, Shankargouda Patil

**Affiliations:** 1Department of Oral Pathology and Microbiology, Faculty of Dental Sciences, Ramaiah University of Applied Sciences, Bengaluru 560054, India; drroopasrao1971@gmail.com (R.S.R.); drsuri29@gmail.com (S.L.); 2Department of Computer Science and Engineering, Faculty of Engineering and Technology, Ramaiah University of Applied Sciences, Bengaluru 560054, India; divyabies@gmail.com (D.B.S.); mkirtishankar@gmail.com (K.S.M.); sinchanagowda28@yahoo.com (S.G.S.); spoorthi.ambika@gmail.com (S.P.); 3Department of Maxillofacial Surgery and Diagnostic Science, Division of Oral Pathology, College of Dentistry, Jazan University, Jazan 45142, Saudi Arabia

**Keywords:** dentigerous cysts, histopathology images, image classification, odontogenic keratocysts, radicular cysts, deep learning

## Abstract

Background: The goal of the study was to create a histopathology image classification automation system that could identify odontogenic keratocysts in hematoxylin and eosin-stained jaw cyst sections. Methods: From 54 odontogenic keratocysts, 23 dentigerous cysts, and 20 radicular cysts, about 2657 microscopic pictures with 400× magnification were obtained. The images were annotated by a pathologist and categorized into epithelium, cystic lumen, and stroma of keratocysts and non-keratocysts. Preprocessing was performed in two steps; the first is data augmentation, as the Deep Learning techniques (DLT) improve their performance with increased data size. Secondly, the epithelial region was selected as the region of interest. Results: Four experiments were conducted using the DLT. In the first, a pre-trained VGG16 was employed to classify after-image augmentation. In the second, DenseNet-169 was implemented for image classification on the augmented images. In the third, DenseNet-169 was trained on the two-step preprocessed images. In the last experiment, two and three results were averaged to obtain an accuracy of 93% on OKC and non-OKC images. Conclusions: The proposed algorithm may fit into the automation system of OKC and non-OKC diagnosis. Utmost care was taken in the manual process of image acquisition (minimum 28–30 images/slide at 40× magnification covering the entire stretch of epithelium and stromal component). Further, there is scope to improve the accuracy rate and make it human bias free by using a whole slide imaging scanner for image acquisition from slides.

## 1. Introduction

Artificial Intelligence and machine learning has evoked interest and opportunities propagating research in health care. The newly developed automated tools that target varied aspects of medical/dental practice have provided a new dimension to translate the laboratory findings into clinical settings [1]. The automated tools act as an adjunct to a pathologist and meet the shortage of experts, which, furthermore, integrates experts of two disciplines, i.e., pathology and computer engineering. Although a pathologist provides a conclusive microscopic diagnosis to clinically challenging lesions, at times a pathologist may go for clinicopathological or radiographic correlation in case of inadequate biopsies [2].

However, if one focuses on the source and output, mere automation of images may not yield the desired results. Thereby, it is necessary to maintain the patient clinical details and follow-up data for a minimum of five years [1].

Pathology is the branch of medicine that deals with the microscopic examination of biopsied tissues for diagnostic purposes. The clinical diagnosis done by medical professionals mandates pathologist’s consultations to tailor the treatment [3]. Thus, histopathology is traditionally considered a gold standard to confirm the clinical diagnosis.

A routine non-digitalized diagnostic pathology workflow involves procurement, preservation, processing, sectioning, and staining of the biopsied tissue to create glass slides followed by an interpretation [3]. The challenging cases are often consulted for multiple expert opinions. Other challenges encountered are ambiguity in diagnosis superimposed with inflammation, inter/intraobserver bias, etc. Although the diagnostic workflow is an exhaustive procedure, automation can ease out the pathologist’s burden by providing a quick and reliable diagnosis [2].

The major Machine Learning challenges to analyze the histopathology images include (1) The requirement of a large dataset to analyze the histopathology images through machine learning algorithms. (2) Identification and assessment of biological structures such as nuclei, with varied shapes and sizes. (3) To detect, analyze, and segment tissue structures in the stroma, such as glands and tumor nests. (4) Lastly, to classify the entire slide image with stroma and epithelial cells [1,4,5].

Literature evidence shows that DL has been applied in analyzing images of major cancers such as breast, colon, and prostate affecting people globally, while rare diseases are seldom addressed by ML tools due to the paucity of data [2,6,7,8]. Furthermore, there are other wider applications of AI models in dentistry that are convolutional neural network (CNN) and artificial neural network (ANN) centric. These AI models have been used to detect and diagnose dental caries, vertical root fractures, apical lesions, salivary gland diseases, maxillary sinusitis, maxillofacial cysts, cervical lymph node metastasis, osteoporosis, alveolar bone loss, and for predicting orthodontic diagnosis and treatment [9,10], genomic studies of head and neck cancer [11], diagnosis and prediction of prognosis in oral cancer [9], and oncology [12,13], etc. Moreover, Majumdar B et al. (2018) highlighted the benefits of AI-based dental education as it can lower the cost of education and ease the strain on educators [14].

Odontogenic keratocysts (OKCs) are relatively rare jaw cysts that account for 3–11% of all jaw cysts^4^. It is found to be the third most common cyst in the Indian population. They are locally aggressive cystic lesions causing bony destruction of the jaws and root resorption of teeth [15,16,17].

A clinical feature that warrants its recognition as a distinct entity is an increased recurrence rate ranging from 2.55–62% and its malignant potential ranging between 0.13% and 2%. The high recurrence is a noted feature of OKCs in patients with nevoid basal cell carcinoma syndrome (NBCCS) [16]. Odontogenic keratocyst (OKC) was studied as a tumor to establish an impact of the reclassification and redefinition on the incidence of odontogenic tumors (OT) [18]. OKC may raise at any age [19].

OKCs have a unique microscopic appearance with 5–8 layers of para or orthokeratinized epithelium and a basal layer with tall columnar/cuboidal cells depicting a typical “tombstone” appearance with polarized nuclei, while other common jaw cysts (non-keratocysts), such as radicular and dentigerous cysts, account for 50% and 20%. Differentiating keratocysts from non-keratocysts is quite challenging with an absence of a unique microscopic appearance. Inflammation further complicates the microscopic evaluation. Rather location, dental procedures opted, or inflammation defines them [2].

The extent of the OKC lesion in the jaw, its aggressive clinical behavior, and high recurrence rate puts the clinicians into a dilemma with respect to therapeutic doctrine. Jaw cysts are frequently observed at dental institutes and are less frequently encountered by pathologists at medical institutes. There are no quantitative criteria in place which can eliminate the subjectivity bias and bring in more objectivity in the microscopic diagnosis of jaw cysts [2].

The treatment of odontogenic keratocysts remains controversial, with surgeons opting for conservative or radical approaches. Orthokeratinised keratocysts are treated less aggressively when compared to parakeratinized keratocysts and the associated syndrome. Clinicians continue to rely on their personal experience to opt for the most appropriate treatment.

Thereby, to resolve these issues, the study aimed to design a histopathology image classification automation system to diagnose and differentiate jaw cysts based on routine hematoxylin and eosin-stained slide images of incisional biopsies. This would deploy ML algorithms. This approach minimizes trauma to the patients and aids the surgeons to plan treatment management.

Here this study considered a relatively large image dataset of 2657 and each class had more than a thousand images, which is the basic requirement of the deep learning algorithm. The images were diverse. Thus, the proposed framework can be integrated into the automatic jaw cysts diagnosis system.

## 2. Materials and Methods

### 2.1. Tissue Specimens

Formalin-fixed (10% buffered) paraffin-embedded biological specimens that correspond to 54 cases of OKCs, 23 cases of DC, and 20 cases of RCs were retrieved from the archives of the Faculty of Dental Sciences, Dept. of Oral pathology, Ramaiah University of Applied Sciences. Next, 4 microns thick sections were cut and stained with hematoxylin and eosin (H&E). The patient’s identity was concealed, while high-resolution images of microscopically confirmed cases of OKCs, RC, and DC were utilized for the present study. This work was approved by the Ethics Committee of Ramaiah University of Applied Sciences (Registry Number EC-20211/F/058).

### 2.2. Image Dataset

The dataset was obtained using Olympus BX53 Research Microscope with a digital Jenoptik camera and Gryphax imaging software. The images of the tissue specimens were of the dimension 3840 × 2160 pixel (px) and are saved in the jpg format.

Manually the images of the H&E-stained section of OKC, DC, and RC were captured at 40× magnification. The consistency of 30 images/slides could not be maintained, because pathological specimens differ from case to case. Furthermore, other factors, such as size, length, epithelial convolutions, and presentation of the pathognomonic features, do matter. This mandate exploring the entire stretch of epithelium. Those specimens with inflammation further bring about certain changes in the epithelium.

The manually obtained images were annotated by an experienced pathologist. Firstly, the jaw cysts were segregated into keratocysts and non-keratocysts employing the standard diagnostic criteria, as the keratocysts present with a distinct histologic appearance, such as parakeratinised squamous lining epithelium comprising of 5–8 layers of cells, while the basal cells are cuboidal or columnar, have elliptical nuclei, and are consistently aligned, resulting in a palisading pattern [20], while the non-keratocysts lack definitive histologic features. Inflamed keratocysts lacking typical epithelium, inadequate biopsies etc. were further classified as challenging ones.

Approximately 1384 images were of OKC and 1273 were images of the non-OKC class (where 636 were images of dentigerous cysts and 637 of radicular cysts). The images covered both the epithelial and the sub-epithelial-stromal components. Only a few cysts were completely devoid of epithelial components, consisting only of the fibrous or inflammatory stroma. 70% of the dataset was used as a training set, 15% as validation, and the remaining 15% as a test set (Figure 1).

### 2.3. Computational Framework

The computation mentioned in the algorithm was performed by using cloud computing environment Google Colab, GPU—Tesla K80, RAM 12 GB, personal computer (Intel^(R)^ Core ^(TM)^ i3-4030U CPU @ 1.90 GHz) CNN was built with Keras.

#### Preprocessing

In the proposed framework preprocessing was one of the critical steps.

Step 1: Data-augmentation

In preprocessing, the dataset was augmented as the accuracy of DL algorithms increased with the increase in the number of images in the dataset. Data augmentation is a technique that assists machine learning programmers in significantly increasing the size of data available for training models.

Image Data Generator class was used for the dataset augmentation. The images size is set to (224, 224).

The data augmentation method is mainly used to get more images in the training dataset, so that we can improve the efficiency of the model and make it more generalized. This data augmentation will also help to overcome the overfitting problem posed by transfer learning. And this augmentation method only applied to the training set, and not on the validation, or test set (Table 1).

The width-shift range and height-shift range arguments are provided to the data generator constructor to adjust the horizontal and vertical shift. For zooming of the images, the argument in the data generator class will take the float value. And the zoom-in operation will be performed when the given value to the argument is smaller than 1 and zoom-out will have performed when the value given is larger than 1. Table 1 shows the details of data augmentation.

Step 2: Region Selection

In the images two regions were majorly observed, the epithelial region and connective tissue region. The epithelial region of OKC had distinct features such as a palisading pattern of basal cells and parakeratinized surface which distinguishes from DC and RC, so here an attempt was made to retain only the epithelial region and remove the connective tissue region. The images were titled into nine patches, each patch of resolution 1280 × 720. The variance was calculated on each patch, the average was calculated over the variance values obtained, and the region with the variance less than the average was marked as connective tissue region. To confirm the connective tissue region, on the same patch, average intensity is calculated after converting into grayscale, then histogram was plotted for each patch. If the histogram had more values for the intensity above the average intensity, then this confirms the patch belongs to connective tissue. The confirmed patches pixel values were made zero. These tiles were concatenated again to get the original resolution (Figure 2).

Region selection would have been achieved through an AI technique, such as semantic segmentation using Region CNN or UNet, where every pixel would be labeled to any of the classes, here, epithelial region and connective tissue region. These techniques needed massive, labeled data. Creating such labeled data, one should use a tool, such as the drawing pen tool of Photoshop or Adobe, to select the region of interest and label the pixels. This process would have been very time-consuming and tedious. Moreover, AI-based region selections, such as region-based convolutional neural networks or U-Net, were computationally expensive and need high-end machines. To make the developed technique usable for the public, these techniques had to be integrated with a desktop application, mobile application, or web application. In this case, the executable code may become too bulky to fit in the application and may take a longer time to execute and show the results. Therefore, in the present research, a very simple, computationally inexpensive, and very light image processing-based region selection technique was used.

### 2.4. Training of Convolutional Neural Networks

The experiment was conducted by training the comparatively simple CNN model VGG16 on the images with step 1 data augmentation and preprocessing.

The VGG model was developed by Simonyan with a very small convolutional in the network, as we know that it is a simple model, and it is a more widely applied model as compared to other models, because of its structure and the association between the convolutional layers. The VGG16 architecture has thirteen convolutional layers in the order Conv 1, Conv 2, Max pool 1, Conv 3, Cpnv 4, Max pool 2, Conv 5, Conv 6, Conv 7, Max pool 3, Conv 8, Conv 9, Conv 10, Max pool 4, Conv 11, Conv 12, Conv 13. This is followed by three fully connected layers with SoftMax as an activation function for the output layer. The Conv 1 and Conv 2 have sixty-four feature maps, which are resulted from sixty-four filters of size 3 × 3. Conv 3 and Conv 4 have a hundred and twenty-eight feature maps, resulting from a hundred and twenty-eight filters of size 3 × 3. Conv 5, Conv 6, and Conv 7 have two hundred and fifty-six feature maps, which are resulted from two hundred and fifty-six filters of size 3 × 3. Conv 8, Conv 9, and Conv 10 have five hundred and twelve feature maps, which are resulted from five hundred and twelve filters of size 3 × 3. Conv 11, Conv 12, and Conv 13 have, again, five hundred and twelve feature maps, which are resulted from five hundred and twelve filters of size 3 × 3. All the convolution layers were built with a one-pixel stride and one pixel of zero paddings. All the four max pooling’s were done on a 2 × 2-pixel window and with stride 2.

Every convolutional layer will follow a ReLU layer and for sampling it has maximum pooling layers. For the classification, it has 3 layers that are fully linked for the classification, in which 2 serve as hidden layers and the last one will be the classification layer. The first layer had 25,088 perceptron’s, the second had 4096, and the third had 2, as we were here performing binary classification of OKC and non-OKC.

Transfer learning is transferring the learned knowledge from a dataset by a network for solving similar kinds of problems on the dataset which has fewer instances in the dataset.

The belief in transfer learning is that the model trained on a huge and generic dataset may suit for classification of the dataset with a smaller number of images. One can use these learned feature maps, instead of training the model from scratch and in transfer learning, we have the privilege that we can consider only part of the model, or full model, as per our problem and we can take those considered part of the model weights to extract specific features from the dataset. Lower layers will be updated as per our classification problem.

In this work for the automation of OKC image classification, an already pre-trained VGG16 model was considered. VGG16 has 16 layers in total; the first thirteen layers are pre-trained on the data set ImageNet, which has nearly 1.2 million training images of 22,000 categories. The required image size for the transfer learning model is VGG16 (224 × 224 × 3). Only the last three layers were trained for the dataset in hand.

### 2.5. Experiment II

Experiment II was conducted by training the effective CNN model DenseNet169 on the preprocessed images with step 1 data augmentation.

Dense Net169 was trained on the given dataset. Dense Net is inspired by the study which showed that convolutional neural networks, which have short connections between the layers closer to the input layer than those which are closer to the output layer, are efficient to train and, at the same time, can grow deeper and have good accuracy. In Dense Net, each layer is connected to the other layer having the same feature-map size in a feed-forward manner. In the case of traditional networks, where the number of connections is equal to the number of layers, in Dense Net, the number of connections is calculated as $\frac{N\left(N+1\right)}{2}$ where N represents the number of layers.

This network architecture allows reusing the feature maps; it improves the information flow in the network through direct connections and reduces the number of parameters. The number of filters for each layer in Dense Net can be as small as 12. These densely connected links provide the effect of regularization which prevents overfitting in such cases where the training data are small.

Let *y*_0_ represent an image, *N* is the number of layers in the network, Cn (.) is the composite function. The index of each layer is represented by n. Then, the input to the last layer is represented by Equation (1).
(1)$$\mathrm{Yn}=\mathrm{Cn}\left(\left[y_{0},y_{1},y_{2}\ldots ,y_{n-1}\right]\right)$$

[*y*_0_,*y*_1_,*y*_2_,….,*y*_*n* − 1_] is obtained by concatenating the feature maps from layers 0 to n − 1. Cn is a composite function of three operations in the order: Batch normalization, ReLU, and 3 × 3 convolution.

The architecture of the Dense Net is divided into several blocks referred to as ‘dense blocks. These blocks are separated by transition layers, which consist of convolution and pooling layers. This CNN captures the overall image features including the unique feature of OKC, separation between epithelium, and connective tissue region. This also helps in capturing the inflamed OKC, where the tombstone arrangement of basal cells was disturbed.

### 2.6. Experiment III

DenseNet169 was trained on the preprocessed dataset. This dataset was created by retaining the patches with epithelium layer in OKC and non-OKC (DC and RC) as explained in the preprocessing section. This CNN is trained to capture the regularity in basal cell arrangement (tombstone arrangement) and 5 to 8 layers of basal cell.

### 2.7. Experiment IV

The models trained in experiments II and III were integrated by averaging the resultant confidence scores to get the predicted output.

The overall architecture of experiment IV is as shown in Figure 3.

The optimizer used is the Adam optimizer, and binary cross entropy is the loss function.

## 3. Results

There were 1384 images of OKC and 1273 images of non-OKC; 15% of the images were used for testing the trained model. In total 207 images of OKC and 191 images of non-OKC were used for testing.

The loss function used in each classifier model discussed here is binary cross entropy. This is most used for binary classification problems. The formula for calculating the binary cross entropy loss or log loss is given by Equation (2).
(2)$$\mathrm{Loss}=-\frac{1}{N\;}\sum_{i=1}^{N}y_{i}.\log\left(p\left(y_{i}\right)\right)+\left(1-y_{i}\right).\log\left(1-p\left(y_{i}\right)\right)$$

Accuracy is the fraction of several predictions done correctly by the model out of the total number of samples. The formula for calculating accuracy is given by Equation (3).
(3)$$\mathrm{Accuracy}=\frac{True\;positive+True\;negative}{True\;positive+True\;negative+False\;positive+False\;negative}$$

Precision is the fraction of several true positive cases out of the number of samples that are predicted positively by the model. The formula for calculating precision is given by Equation (4).
(4)$$\mathrm{Precision}=\frac{True\;positive}{\;true\;positive+False\;positive}$$

Recall is the fraction of many true positive cases out of the number of actual positive cases. The formula for calculating recall is given by Equation (5).
(5)$$\mathrm{Recall}=\frac{True\;positive}{\;true\;positive+False\;Negative}$$

The F1-score is the harmonic mean of precision and recall. The formula for calculating the F1-score is given by Equation (6).
(6)$$F1-\mathrm{score}=2*\frac{Precision*Recall}{Precision+Recall}$$

ROC or Receiver Operator Characteristic curve is a graph that plots the true positive rate against the false-positive rates at different threshold values. This is particularly useful in binary classification problems.

A confusion matrix is a table that gives us a summary of the model’s performance. The format of the confusion matrix for a binary classification problem is shown in (Table 2).

The macro-average gives the overall performance of the classifier. The macro average is the arithmetic mean of individual classes’ precision, recall, and F1-score.

Weighted avg gives the function to compute precision, recall, and F1-score for each label and returns the average considering each label’s proportion in the dataset.

### 3.1. The Training Phase

In the training phase the parameters given to the developed model were: 

(a)‘Adam’ optimizer

An optimization algorithm plays an important role in deep learning algorithms, as it is a strategy that is performed iteratively until an optimum solution is obtained. Adam optimizer is a hybrid of Adagrad and RMSProp algorithms to produce an optimum solution for a given problem.

(b)minimum batch size

Updating the internal model parameters would be tedious if done after every sample, so samples are grouped as batches and the model parameter is updated for these batches.

Batch size is a hyperparameter. Here, 11 histopathological images were grouped as batches.

(c)the number of training epochs

One epoch means the entire training dataset was used to update the internal model parameters once. The number of epochs is a hyperparameter.

(d)initial learning rate

The learning rate is also a hyperparameter. In the case of stochastic gradient descent optimization algorithm learning rate is the amount of the internal model parameter to change concerning the calculated error.

### 3.2. Results Experiment I

Hyperparameters such as the number of epochs and batch size are decided based on experimentation and comparing the results.

In experiment I the VGG16 model performed its best when trained for nine epochs with 12 as the batch size. At the end of nine epochs, the validation accuracy is 89.01% and test accuracy is 62% as shown in Figure 4.

### 3.3. Experiment II

In experiment II, only step 1 data augmentation was performed in preprocessing. That is, the CNN was trained only on the entire image.

In experiment II, the DenseNet169 model performed its best when trained for 15 epochs with 12 as the batch size. At the end of 15 epochs, the validation accuracy is 89.82% and test accuracy is 91%. The AUC for this model is 95.966%.

The plot of accuracy on training and validation data is as shown in Figure 5B. From this plot, we can find that the accuracy of the model for training and validation data converged and is stable at the end of 15 epochs, which indicates that the model is not overtrained and, also, did not overfit. The plot of loss on training and validation data is as shown in Figure 5B. From this plot, we can find that the loss of the model for training and validation data reduced as the training progressed. It converged after a few epochs and became stable later on. The confusion matrix and classification report are shown in Figure 5A. From the confusion matrix, we can understand that the number of true positives for this model is 181 out of 207 positive samples. The ROC curve is shown in Figure 5C.

### 3.4. Experiment III

In experiment III, both the steps were performed in preprocessing. That is, the CNN was trained only on the features of the epithelium.

On experimentation and comparison, Model-3 gave its best predictions when trained for 100 epochs with 16 as the batch size. At the end of 100 epochs, the test accuracy is noted as 91%. The AUC for this model is 96.375%.

The plot of accuracy on training and validation data is as shown in (Figure 6B). From this graph, we can observe that the validation accuracy is varying throughout the training, but, in the end, it stabilizes. The gap between the two lines shows some over-fitting on the validation data.

The plot of loss on training and validation data is as shown in (Figure 6B). From this graph, we can find that the training loss reduces in the first few epochs and then becomes completely stable. Similarly, the validation loss is almost stable with very little variation throughout the training (Figure 6B). The confusion matrix and classification report are shown in Figure 6A. On observing the confusion matrix, we can find that there are 191 correctly identified positive samples out of 207 positive samples. The ROC curve is as shown in Figure 6C.

### 3.5. Experiment IV

In experiment IV, the models trained in experiments II and III were integrated by averaging the resultant confidence scores to get the predicted output. The confusion matrix and classification report are shown in Figure 7A. The confusion matrix shows that this combined architecture has correctly identified 189 samples out of 207 samples belonging to the true class. The ROC curve is as shown in Figure 7B.

## 4. Discussion

The main features to look at in the OKC histopathology image pattern are thick epithelium, 5–6 layers of a regular arrangement of basal cells, also called tombstone arrangement, and the separation of epithelium and the connective tissue. On the other hand, DC has a thin epithelium layer with 2–3 layers of irregularly arranged basal cells and the epithelium layer is not properly distinguishable and has penetration into connective tissue in RC. The classification is more challenging when they undergo inflammation, the epithelium layer of DC becomes thick, which may be confused for OKC.

These images show large diversity within the class, posing a challenge for automation. Deep learning has shown promising results in the automation of digital histopathological image classification. Automation of digital histopathological image classification brings the standardization in the procedure, by eliminating the manual observation of the tissues under a microscope by the pathologists, which is completely subjective and suffers inter-laboratory variations.

In the proposed design, during preprocessing an attempt was made to select the epithelial region in the images. A Dense Net CNN was trained on these datasets to identify the features in the epithelium which distinguishes OKC from RC and DC. Another Dense Net was trained on the whole image to capture the separation of epithelium and the connective tissue feature. As the dataset had a limited number of images, transfer learning was adopted. The transfer learning approach results in better accuracy and reduced training time. Transfer learning is reusing a pre-built and pre-trained model on a new dataset. With transfer learning, we can reduce training time and get the best results even when many of data are unavailable. Both Dense Net CNNs obtained 91% accuracy over the dataset. An ensemble model of the Dense Net CNNs obtained an accuracy of 93%. Densenet169 was pre-trained on the ImageNet dataset only the last layer was trained for the given dataset. The problem of the OKC and non-OKC classification was addressed by using two CNN models. First, CNN was trained on the hand-crafted small patches of epithelium and the second on the whole image. They obtained a higher accuracy of 98% as they had a large dataset of 1704 OKC and 1635 non-OKC images [21].

The classification of periapical cysts (PCs), dentigerous cysts (DCs), ameloblastomas (ABs), odontogenic keratocysts (OKCs), and normal jaws with no diseases, considering the dataset for panoramic radiographs, were also attempted [22]. The classification performance of CNN for PCs, DCs, ABs, OKCs, and normal jaws for sensitivities, specificities, accuracies, and AUCs are: 82.8%, 99.2%, 96.2%, and 0.92 (PCs), 91.4%, 99.2%, 97.8%, and 0.96 (DCs), 71.7%, 100%, 94.3%, and 0.86 (ABs), 98.4%, 92.3%, 94.0%, and 0.97 (OKCs), and 100.0%, 95.1%, 96.0%, and 0.94 (normal jaws), respectively. The work process for assisting the dentist was analyzed, with emphasis on the automatic study of the cyst using texture analysis [23,24]. Keratocystic odontogenic tumor diagnosis automation is in the infancy stage [22]. A survey stated that feature retrieval for CNN accomplished fine-tuning in image classification [15]. The occurrence of several OKCs is one of the chief conditions for the analysis of nevoid basal cell carcinoma syndrome [25]. OKC lesions were commonly found more in females than males [26]. The recurrence of OKCs was in between five to seven years, but recurrence in the range of 12 to 102 months was also reported [27,28]. The management of OKCs did not accept any protocol [29]. An approach was proposed, known as the Bouligand–Minkowski descriptors (B–M), to evaluate the success rates based on the epithelial lining classification of these cysts using a histological image database [30].

The current investigation used H&E-stained sections on incisional biopsies, which are globally acknowledged, cost-effective, and time-tested. However, there are other ways to reduce the dataset by opting for immunohistochemical (IHC) staining specific to the nucleus. In the absence of a typical epithelium, in case of inadequate biopsy, the presence of non-keratinizing epithelium with basal palisading and an immunophenotype characteristic of OKC (basal bcl2, patchy or diffuse CK17, and upper layer CK10 positivity) may be consistent with the OKC diagnosis [31]. A basic nuclear staining approach like DAPI staining can be used to detect palisading patterns [2].

## 5. Conclusions

The hematoxylin and eosin-stained tissue specimens of OKC and NK were collected as a dataset. Two convolutional neural network models were trained on the region-selected dataset and the whole image dataset separately. These were ensemble by averaging their confidence scores, to give better accuracy. This architecture could be computationally expensive and may require a faster CPU to overcome. If IHC is opted for as a choice to reduce the dataset, utmost care must be taken on the economic viability while using IHC.

## Figures and Tables

**Figure 1 diagnostics-11-02184-f001:**
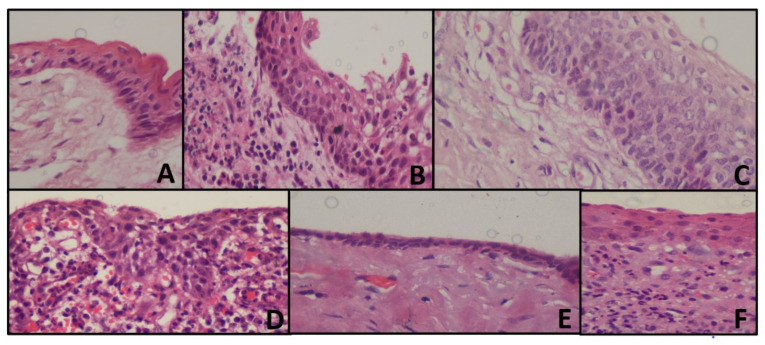
Representing histopathology images of the cyst (**A**) OKC with tombstone appearance of basal cells, corrugated epithelium without inflammation (**B**) Loss of classic appearance of OKC with underlying inflammation (**C**) Tombstone appearance, without corrugation with reversed polarity (**D**) Radicular cyst showing arcading pattern with inflammation (**E**) Dentigerous cyst, showing cystic lining without inflammation (**F**) Dentigerous cyst, showing inflammatory connective tissue.

**Figure 2 diagnostics-11-02184-f002:**
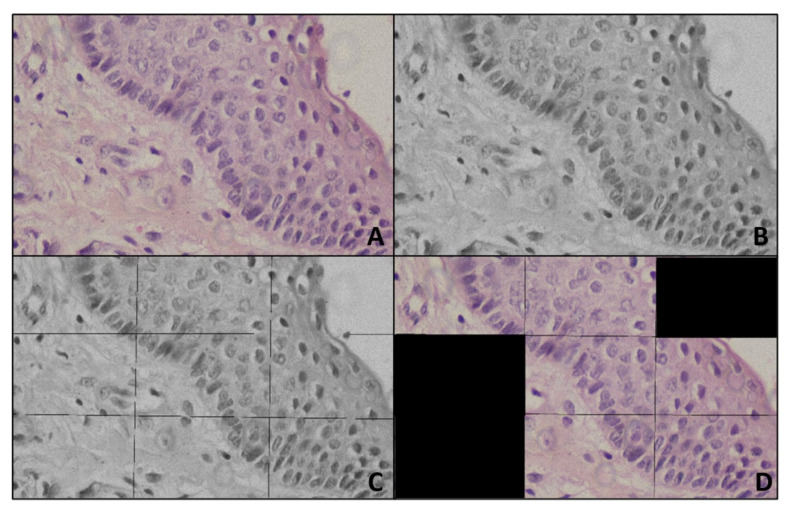
Representing preprocessing (**A**) Input image (**B**) Gray image (**C**) Titled gray image (**D**) Output of Preprocessing.

**Figure 3 diagnostics-11-02184-f003:**
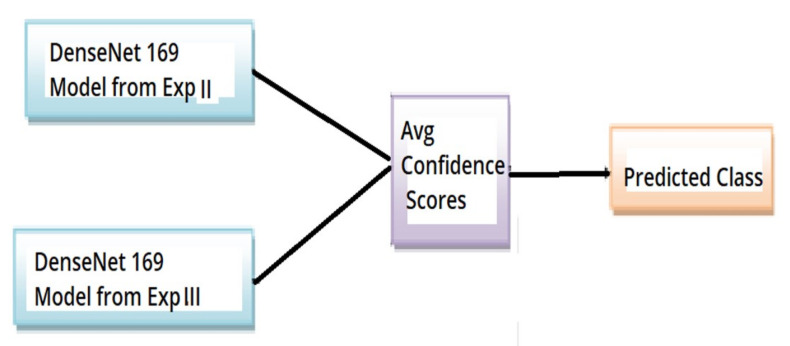
Overall architecture of experiment IV OKC classifier.

**Figure 4 diagnostics-11-02184-f004:**
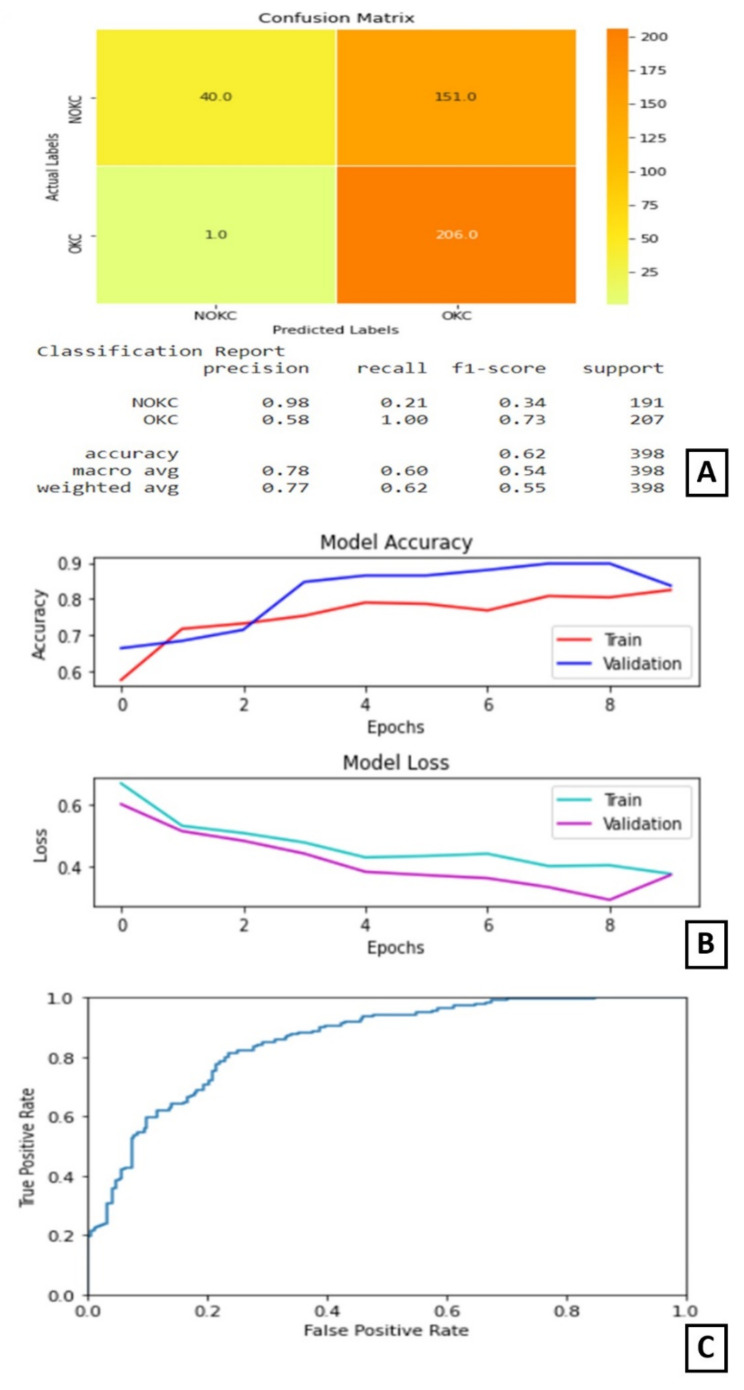
Results of Experiment I (**A**) Confusion matrix and performance metric (**B**) Training parameters (**C**) ROC curve.

**Figure 5 diagnostics-11-02184-f005:**
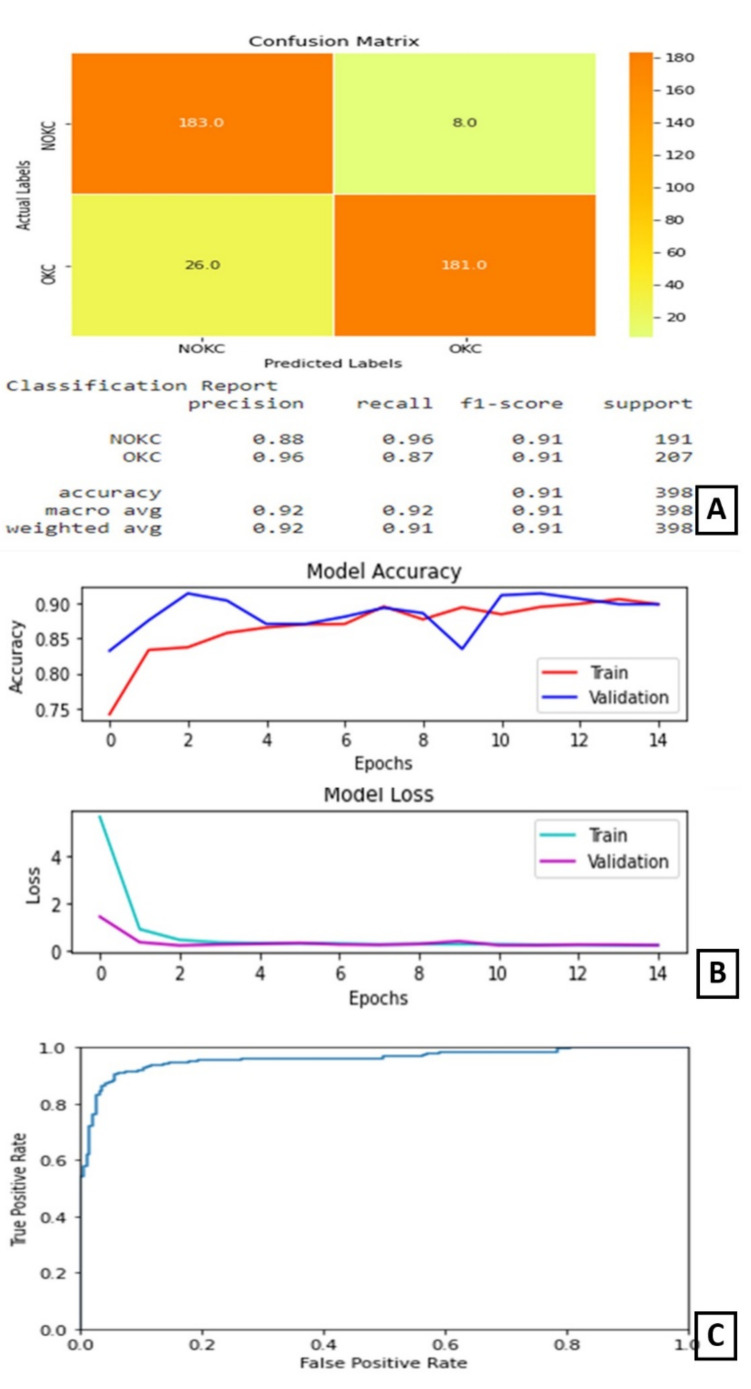
Results of Experiment II (**A**) Confusion matrix and performance metric (**B**) Training parameters (**C**) ROC curve.

**Figure 6 diagnostics-11-02184-f006:**
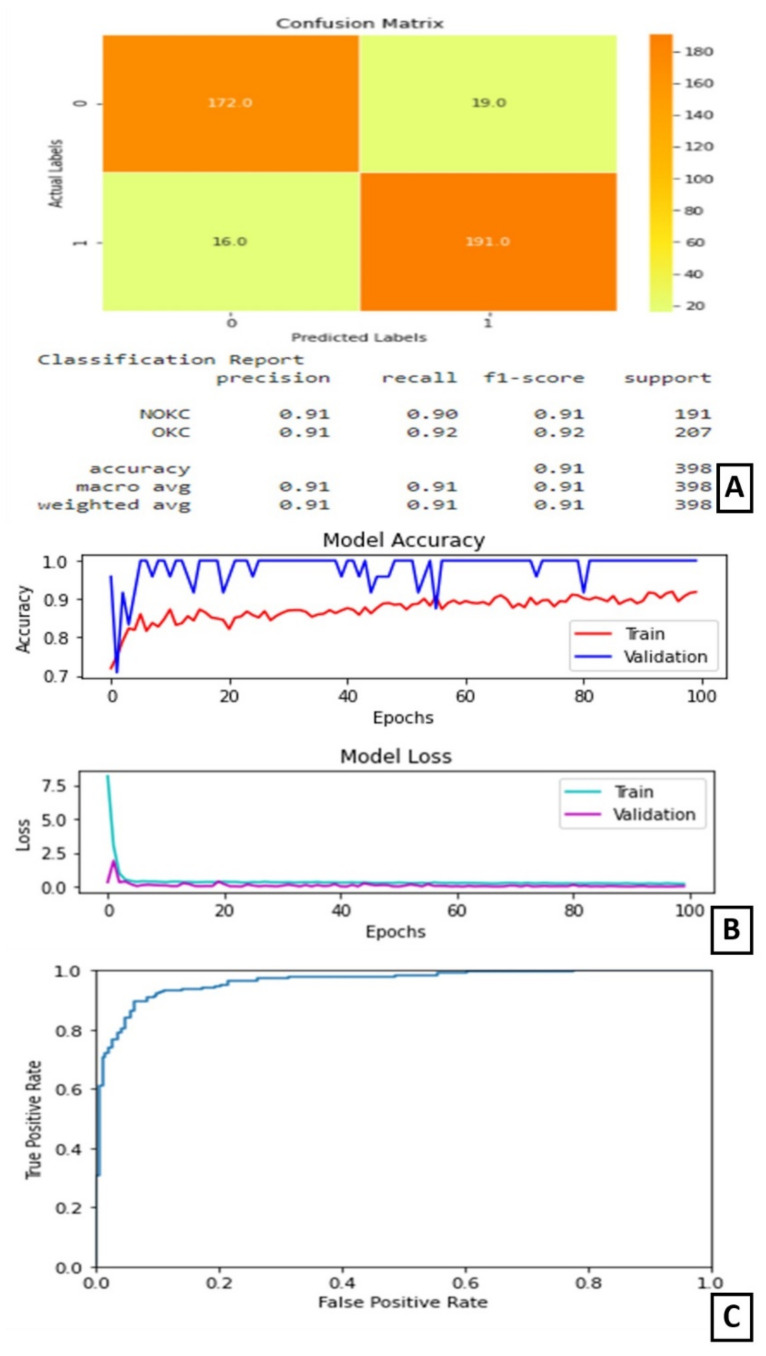
Results of Experiment III (**A**) Confusion matrix and performance metric (**B**) Training parameters (**C**) ROC curve.

**Figure 7 diagnostics-11-02184-f007:**
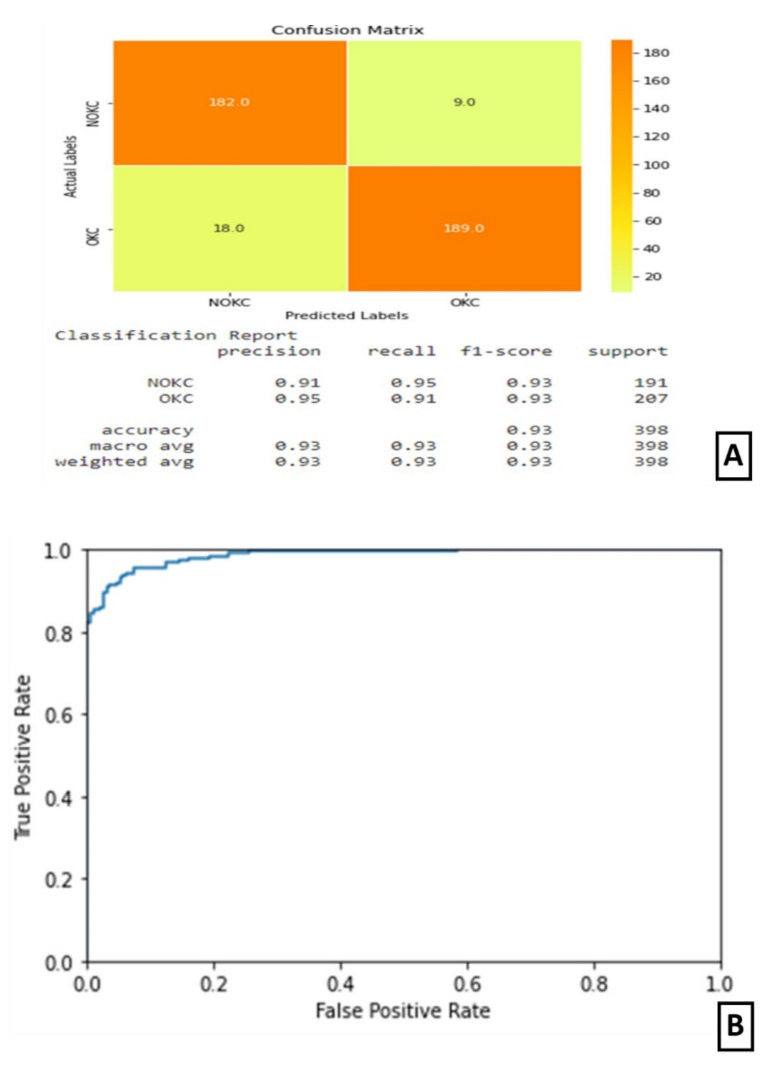
Results of Experiment IV (**A**) Confusion matrix and performance metric (**B**) ROC curve.

**Table 1 diagnostics-11-02184-t001:** Details of data augmentation Techniques.

Data Augmentation Technique	Value
Shear range	0.2
Rotation range	20
Horizontal flip	True
Vertical flip	False
Zoom range	0.5

The data augmentation methods used were as follows: Image shear is a bounding box transformation. Rotation of the image will be done by the rotation range. Image flipping is done by the horizontal flip and vertical flip. Zooming of the images is done by the zoom range.

**Table 2 diagnostics-11-02184-t002:** Description of the confusion matrix.

	Actual Values	
Predicted Values	True positive	False-positive
	True negative	False-negative

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
