# Peer review of "Deep Learning-Based Microscopic Diagnosis of Odontogenic Keratocysts and Non-Keratocysts in Haematoxylin and Eosin-Stained Incisional Biopsies"

_diagnostics, 2021, doi:10.3390/diagnostics11122184_

Round 1

Reviewer 1 Report

The experiments are described quite thoroughly; however, it is not clear what criteria were used by pathologists in the manual process of image acquisition from slides. I wonder if the method is human bias free. It would be better to use slides full scans to avoid any human-based influence.

Also, the research could be augmented by the replacement of a simple image processing algorithm for epithelial region selection by the modern AI approach. Image pixel-level segmentation would improve region selection accuracy and perhaps, enhance the accuracy of the entire method. However, Ia am aware that this requires tedious image labeling process.

Also, it is not clear to me why you mention "X-ray radiograph" images in the batch size-related paragraph.

Author Response

The experiments are described quite thoroughly; however, it is not clear what criteria were used by pathologists in the manual process of image acquisition from slides. I wonder if the method is human bias free. It would be better to use slides full scans to avoid any human-based influence.

[30 -33] – Utmost care was taken in the manual process of image acquisition (minimum 28-30 images /slide at 40x magnification covering the entire stretch of epithelium and stromal component. Further, there is scope to improve the accuracy rate and make it human bias free by using a whole slide imaging scanner for image acquisition from slides.

 This is a pilot study generated manually. It is approved by SERB power funding , where we can procure whole slide imaging scanner .

Also, the research could be augmented by the replacement of a simple image processing algorithm for epithelial region selection by the modern AI approach. Image pixel-level segmentation would improve region selection accuracy and perhaps, enhance the accuracy of the entire method. However, Ia am aware that this requires tedious image labeling process.

[206-218]- Region selection would have been achieved through an AI technique, like semantic segmentation using Region CNN or UNet, where every pixel would be labelled to any of the classes, here epithelial region, and connective tissue region. These techniques needed massive, labelled data. Creating such labelled data, one should use a tool like the drawing pen tool of Photoshop or Adobe to select the region of interest and label the pixels. This process would have been very time-consuming and tedious. Moreover, AI-based region selection like region-based Convolutional Neural networks or U-Net were computationally expensive and need high-end machines. To make the developed technique usable for the public, these techniques had to be integrated with a desktop application or mobile application, or web application. In this case, the executable code may become too bulky to fit in the application and may take a longer time to execute and show the results. So, in the present research, a very simple, computationally inexpensive, and very light image processing-based region selection technique was used.

Also, it is not clear to me why you mention "X-ray radiograph" images in the batch size-related paragraph.

[378] - . Here 11 histopathological images were grouped as batches. It was wrongly written by the computer engineer.

Reviewer 2 Report

The authors aimed to create a histopathology image classification automation system that could identify odontogenic keratocysts in hematoxylin and eosin-stained jaw cyst sections. So, the final scope was that the proposed framework can be integrated into the automatic jaw cysts diagnosis system.

The study covers some issues that have been overlooked in other similar topics. The structure of the manuscript appears adequate and well divided in the sections. Moreover, the study is easy to follow, but some issues should be improved. The manuscript needs moderate grammar correction. Please also check typos thorough the text.

Conclusion Section: This paragraph required a general revision to eliminate redundant sentences and to add some "take-home message".

Author Response

The authors aimed to create a histopathology image classification automation system that could identify odontogenic keratocysts in hematoxylin and eosin-stained jaw cyst sections. So, the final scope was that the proposed framework can be integrated into the automatic jaw cysts diagnosis system.

Yes, the final scope of the proposed framework is, it can be integrated into the automatic jaw cysts diagnosis system.

To be specific, we have used only two cysts of developmental origin and one of inflammatory origin (not covered all of them ) . Thereby categorised into keratocysts and non-keratocysts.

The study covers some issues that have been overlooked in other similar topics. The structure of the manuscript appears adequate and well divided in the sections. Moreover, the study is easy to follow, but some issues should be improved. The manuscript needs moderate grammar correction. Please also check typos thorough the text.

Conclusion Section: This paragraph required a general revision to eliminate redundant sentences and to add some "take-home message".

[516-523] specific IHC markers have been added rather than making a generic statement.

[525-531] Paraphrased the sentences in the conclusion with the take home message as suggested.
